# Post-Separation Physical Custody Arrangements in Germany: Examining Sociodemographic Correlates, Parental Coparenting, and Child Adjustment

Alexandra N. Langmeyer *, Claudia Recksiedler, Christine Entleitner-Phleps and Sabine Walper

German Youth Institute, Nockherstrasse 2, 81541 Munich, Germany; recksiedler@dji.de (C.R.); entleitner-phleps@dji.de (C.E.-P.); walper@dji.de (S.W.)
* Correspondence: langmeyer@dji.de

**Abstract:** Most children continue to live with their mother after a divorce or separation, yet paternal involvement in post-separation families has increased substantially in many Western nations. This shift has contributed to a growing share and more diverse set of post-separation parents opting for shared physical custody (SPC), which typically means that children alternate between the parental residences for substantive amounts of time. Profiling the case of Germany, where no legal regulations facilitating SPC are implemented to date, we examine the prevalence of SPC families, sociodemographic correlates of SPC, and its associations with parental coparenting and child adjustment. Using representative survey data sampled in 2019 ($N$ = 800 minors of 509 separated parents), results revealed that only 6–8% of children practiced SPC. SPC parents were more likely to hold tertiary levels of schooling and to report a better coparenting relationship with the other parent. There was no link between SPC and child adjustment, yet conflictual coparenting was linked to higher levels of hyperactivity among SPC children. We conclude that the social selection into SPC and linkages between conflictual coparenting and hyperactivity among SPC children likely stem from the higher costs and the constant level of communication between the ex-partners that SPC requires.

**Keywords:** shared physical custody; sole care; union dissolution; child wellbeing; parental relationship

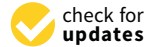



## 1. Introduction

Shared physical custody (SPC) after separation and divorce has become a widely debated issue, not only in family law, but also in social science research in many countries (Walper et al. 2021). Given the intensive debate about the pros and cons of shared post-separation care, a large number of studies focuses on the outcomes of SPC compared to more traditional sole physical custody arrangements, particularly with regard to children's well-being (e.g., Nielsen 2018; Baude et al. 2016; Hjern et al. 2021b). In comparison, the number of studies that addressed issues of selectivity into post-separation physical custody arrangements are more limited (Poortman and van Gaalen 2017; Sodermans et al. 2013; Recksiedler and Bernardi 2021). Nonetheless, both lines of research are similarly important and even mutually dependent, especially because questions concerning the role of custody in children's well-being cannot be adequately answered without considering the selective use of SPC (Fehlberg et al. 2011). Thus, this study takes a look at both aspects: factors associated with the parents' choice of and selection into different physical custody arrangements, as well as the linkages of such arrangements and child adjustment among a sample of German post-separation families.

### 1.1. Physical Custody Arrangements in Separated Families

Since fathers became more involved in family matters and childcare in general (Schoppe-Sullivan and Fagan 2020), the involvement of fathers after separation and divorce also gained importance. The majority of children, especially in western societies, still live

with their mothers after the separation of their parents (Bernardi et al. 2018). However, the share of families practicing SPC, where children live with both parents for approximately equal amounts of time, is increasing, and sole care is becoming less common (Bjarnason and Arnarsson 2011; Meyer et al. 2017). Definitions of SPC mostly range between time splits of 50:50 (in terms of equal shares of overnight stays) to more asymmetrical distributions of overnight stays between parents of up to 70:30 (Baude et al. 2016).

SPC is particularly prevalent in Sweden, with a share of around 30% of children whose parents are separated. This high proportion is possibly due to the fact that Sweden supports more egalitarian gender roles in the family system and allows family courts to order SPC in cases of post-separation legal conflict (Vanassche et al. 2017; Walper et al. 2021). Reforms to family law in Australia (Fehlberg et al. 2011) and Belgium (Sodermans et al. 2013) in the first decade of the new millennium have also implemented and strengthened SPC as the legal norm. This has contributed to an increase in SPC, as well as changes in the conditions under which separated families realize SPC. However, the trend toward increased use of SPC is not always linear: In the Netherlands, SPC increased before and shortly after a reform to family law in 2009, which strengthened SPC. Starting from a low level of about 5% of divorced families in the 1980s and 1990s, SPC increased among recently divorced couples to about 20% between 2000 and 2007 (prior to the reform). In 2009 (post-reform) the share of children in SPC rose to 32%, but decreased in the following years to 22% in 2013 (Poortman and van Gaalen 2017).

In Germany, SPC is not yet established by law, and institutional support for SPC is sparse. It is therefore important to note that findings drawn from Germany may only apply to other countries and jurisdictional contexts with similarly low legal and institutional support for SPC. Due to the lack of legal and institutional support for SPC, its prevalence remains quite low in Germany (Bjarnason and Arnarsson 2011) and lags far behind other European countries with broader support for SPC, such as the Netherlands, England, or Sweden (Kalmijn 2016). However, estimates of the prevalence of SPC in Germany differ. Analyses of the German family panel study pairfam revealed that less than 5% of separated families practice SPC in 2014 (Walper 2016). This finding is in line with estimates from the 2015 installment of the representative, cross-sectional survey "Growing Up in Germany", where only 3% of children in separated families practiced SPC (using a 60:40 time split). Among children living in primarily sole care arrangements, about 21% of the children reported having no contact with the father and somewhat less than half of the children had at least weekly contact with their non-residential father (48%). A good quarter (about 27%) had infrequent contact with the non-residential father (Walper et al. 2021). Köppen et al. (2020) reported a slightly higher prevalence of SPC (about 12%) for Germany. However, a different definition of SPC, namely parents' subjective perception of physical custody arrangements, rather than the exact number of overnight stays, was used in this study.

## 1.2. Factors Associated with Social Selection into Different Physical Custody Arrangements

Prior evidence suggests that separated parents' choice of post-separation physical custody is linked to resources and barriers at the contextual, family, and individual levels (for an overview see Walper et al. 2021).

### 1.2.1. Factors at the Contextual Level

Contextual factors have an influence on the choice of custody arrangements after separation. For instance, legal and institutional conditions seem to affect how many and what kind of post-separation families opt into SPC (Walper et al. 2021). This can be explained by the impact of social institutions on individuals' life courses, which is a key hallmark of the *life course perspective* (Elder and Shanahan 2006). More specifically, social institutions and regulations, such as the respective family law in a given country, channel and guide individuals' behaviors (Mills and Blossfeld 2009). Although individuals and families also have the capability to act agentically and choose what is best for them, their choices tend to be embedded and restricted by the existing set of rules and institutional

support systems for certain behaviors and transitions. Accordingly, countries that have a strong legal foundation for SPC, or where SPC was introduced as the legal default, tend to have a higher share and a more diverse set of families opting into SPC. In Belgium, for instance, the more privileged position of SPC families in terms of SES faded after the introduction of SPC as the legal, court-mandated default for separated parents (Sodermans et al. 2013). In Sweden, where legal regulations also favor SPC, the social acceptance of SPC was also rather high among parents, even if the parental relationship was more conflictual (Fransson et al. 2016), which indicates that SPC has become normative in this context. In countries with little institutional support for SPC, such as in Germany where no legal regulations for SPC exist to date, establishing and negotiating SPC requires parents to be more proactive and invested. Instead, institutional arrangements and the family law of Germany's welfare state, which operate under a more traditional male breadwinner model (Grunow et al. 2018), tend to reinforce the formation of female-headed sole care custody arrangements among post-separation families.

### 1.2.2. Factors at the Family Level

Among factors at the family level, the distance between the parents' homes has shown to be linked to the choice of post-separation physical arrangements, particularly because SPC involves a certain logistical effort. If the distance between the parental homes is longer, it is more difficult to practice SPC (e.g., ensuring that the child is cared for regularly, manages his or her way to school, and sees his or her friends). Thus, several studies showed that SPC is more likely when the parental homes are in close proximity (Kaspiew et al. 2009; Walper et al. 2021; Schier and Hubert 2015; Baude et al. 2016).

Parents' ability to cooperate and work together on parenting issues after the separation, which is usually referred to as coparenting, is also a key factor for parents' choice of a given post-separation physical custody arrangement. Even though SPC is theoretically possible without any exchanges between parents (Kaspiew et al. 2009), it can be assumed that SPC requires at least a certain degree of willingness to cooperate (e.g., when discussing and negotiating the child's schedule). Against this backdrop, findings from a prospective Dutch study showed that pre-divorce interparental conflicts and conflicts during the divorce proceedings had negative effects on the likelihood to opt for SPC (Poortman and van Gaalen 2017). Other studies also indicated that SPC parents had fewer conflicts than parents in sole care models (Cashmore et al. 2010). Walper et al. (2021) also showed that a more positive coparenting relationship between parents was associated with a higher chance to practice SPC in Germany. A more conflictual coparenting relationship, in contrast, was not associated with a higher likelihood to practice SPC or sole care. However, some studies also reported that there are indeed parents who practice SPC despite the presence of interparental conflicts (Kaspiew et al. 2009). It is also possible that these conflicts stem from the greater need for exchange between the parents.

### 1.2.3. Factors at the Individual Level: Age and Gender of the Children, Families' Socio-Economic Conditions, Level of Education and Working Situation of the Parents

Whereas parents of infants and toddlers are less likely to choose SPC, this arrangement is most commonly practiced by families with children aged 3 to 12 years and especially among those with children in elementary school (Juby et al. 2005; Sodermans et al. 2013). In adolescence, the number of SPC children is lower (Spruijt and Duindam 2009). This could be the case because the importance of peers and friends is increasing during this developmental period and the adolescents may not have equal access to their friend groups at both parental homes (e.g., if one parent lives further away). A representative German study with data from mothers reporting on 1090 children also showed that the probability of practicing SPC was highest among children aged 6 to 10 years (Walper et al. 2021). In addition, the age of the children at the time of separation seems to be relevant: A Canadian study indicated that children were more likely to practice SPC if they experienced parental divorce during their adolescence (Juby et al. 2005). To date, studies have rarely considered

the sex of the child, with only one study showing that SPC was more likely for boys (Kalmijn 2016).

Parents' higher socio-economic status (SES) in terms of education and income has also been found to increase the likelihood of SPC (Juby et al. 2005; Spruijt and Duindam 2009; Kaspiew et al. 2009; Walper et al. 2021; Nielsen 2013; Cancian et al. 2014; Kalmijn 2016; Recksiedler and Bernardi 2021). This probably reflects the higher financial demands resulting from SPC, which requires adequate living conditions and child-related equipment in both homes (Melli and Brown 2008). Additionally, it may be easier for mothers to reconcile family responsibilities and employment in SPC arrangements (Bonnet et al. 2018). Yet evidence on maternal employment in the context of SPC is still unclear. On the one hand, SPC could make it easier for mothers to work, but on the other hand, SPC may also be more preferred by working mothers because it makes work duties and childcare more compatible. Accordingly, Kalmijn (2016) showed with the CILS4EU data that SPC was more common among employed mothers in England, Sweden, and Germany. In German survey data, however, no links between maternal employment and post-separation custody arrangements emerged (Walper et al. 2021).

A prospective study from the Netherlands found that parents' level of education, but not their pre-divorce income, was linked to parents' chance to opt for SPC (Poortman and van Gaalen 2017). Additionally, families were more likely to practice SPC if mothers worked more hours before the parental separation. In another prospective study from Canada (Juby et al. 2005), however, there was no independent effect of pre-divorce income on the likelihood of practicing SPC. Interestingly, this study found higher rates of SPC, not only among parents with university-level education, but also among those without a high school diploma only. Moreover, employment conditions were also found to be essential, regardless of income and parental education. SPC was more likely if mothers worked at least part-time before the parental divorce and if fathers did not work in the evenings or on weekends (Juby et al. 2005). Furthermore, flexible and family-friendly working conditions were associated with a higher likelihood to practice SPC, probably because both parents are facing challenges to balance family tasks and employment (Nielsen 2013). Overall, the importance of parents' education and pre-divorce employment conditions seems to be clearer, while the findings on income are still mixed.

### 1.3. Physical Custody Arrangements and Child Adjustment

Going through a separation or divorce represents a stressful life transition that has far-reaching ripple effects on parents' life courses because it can diminish their physical, mental, and often financial well-being (Amato 2010; Cooper et al. 2009; Nomaguchi and Milkie 2020; Raley and Sweeney 2020). At the same time, role transitions that alter family configurations and dynamics also affect the well-being and adjustment of other family members, particularly children (e.g., Lamela et al. 2016). This latter point is outlined in conceptual frameworks such as the *linked lives principle* of the life course perspective (Elder and Shanahan 2006; Settersten 2015), which highlights the interconnectedness and social influence among family members across the life course (Thomas et al. 2017). For example, Choi and Becher (2018) showed that maternal depression, which can be more prevalent after separation or divorce (Amato 2010), was linked to an increased likelihood of child behavioral problems through the mothers' use of harsher parenting practices. It is therefore not surprising that a large number of studies examined the link between child well-being and different post-separation physical custody arrangements while controlling for factors associated with social selection into certain physical custody arrangements (e.g., parental SES, housing distance, children's age, and gender). It is often argued that SPC children benefit from the opportunity to continue having close relationships with both parents (Augustijn 2021a) because SPC children are more likely to maintain closer ties with their fathers compared to children in traditional (female-headed) sole care models (Bjarnason and Arnarsson 2011; Nielsen 2011, 2014; Bastaits and Pasteels 2019). However, the advantage to having better father-child communication in SPC families compared to

families where children are living with a single mother was only observed in some countries (Bjarnason and Arnarsson 2011). It was particularly pronounced in Germany, Italy, and the Netherlands, for example, while it was significantly lower in the U.S. and Belgium. This may indicate that if SPC is used by broader and less selective segments of the population, particularly in cases of contentious and highly conflictual separations (e.g., in the U.S. and Belgium), the SPC-related benefits for the father-child relationship may fade (Sodermans et al. 2013).

In particular, the emotional and behavioral development of the SPC children was studied intensively, and prior studies identified some advantages of this physical custody arrangement (Bauserman 2002; Breivik and Olweus 2006; Turunen et al. 2017; Turunen 2017; Braver and Votruba 2018). Studies from Sweden and Norway, where SPC was introduced rather early and is adopted by a substantial share of post-separation families, suggest that adolescents practicing SPC and those in two-parent families showed comparable levels of well-being and psychosocial adjustment (even though SPC adolescents scored a slightly lower), whereas well-being and adjustment detriments were observable for adolescents living with a single parent (Bergström et al. 2015; Carlsund et al. 2013; Breivik and Olweus 2006; Turunen 2017). This was shown, for example, based on analyses of the Health Behaviour in School-aged Children (HBSC) data for 11- to 15-year-old schoolchildren in Sweden, which focused on examining differences in adolescents' subjective well-being and health impairments (Carlsund et al. 2013). Children and adolescents in two-parent families reported experiencing less stress and fewer health impairments compared to their peers living in SPC or sole care arrangements. However, health and well-being detriments were stronger for children in sole care models compared to SPC children and these differences remained highly salient when controlling for relevant socioeconomic factors and children's relationship quality with both parents. This finding is also supported by a recent longitudinal study from Sweden that covered a time span of over 11 years (Hjern et al. 2021b). It could be shown that children living in two-parent families had the most favorable scores on the Strength and Difficulties Questionnaire (SDQ, Goodman 2006), which is a validated and widely used measure of child maladjustment and behavioral problems. Among children who experienced parental separation or divorce, SPC children had a lower level of mental health problems compared to children in sole care models.

A Norwegian study suggests, in contrast, that adolescents in SPC arrangements did not differ from peers in two-parent families in terms of their level of behavioral problems, especially with regard to internalizing problems (e.g., as depression and a negative self-concept; Breivik and Olweus 2006). Another Swedish study with a broader age range (4–18 years) that assessed children's overall mental health using the SDQ (Goodman 2006), did not find evidence for any clear advantages of SPC children and adolescents' mental health. The mental health disparities that minors in SPC and sole care models displayed (compared to their peers in two-parent families) could largely be explained by the lower levels of parental life satisfaction in both post-separation custody care arrangements (Bergström et al. 2014; Nomaguchi and Milkie 2020). An analysis of the Danish National Birth Cohort Study, with data from 39,661 mothers reporting on their 7-year-old children, also found no differences in the SDQ problem score between two-parent families, SPC, sole care models with and without a new partner in the maternal household (Hjern et al. 2021a). A German study based on a convenience sample (Augustijn 2021a) also showed similar findings: Although SPC children displayed fewer psychosomatic problems than children in sole care models, these differences can at least partly be explained by the selectivity into SPC (e.g., a better parent-child relationship, interestingly especially with the mother rather than the father). Other analyses based on this data also demonstrated that SPC children were better off with regard to other dimensions of well-being (e.g., physical health or school grades) compared to children in sole care models (Steinbach and Augustijn 2021). However, when several control variables were taken into account (e.g., the quality of family relationships), these differences disappeared. This indicates again that parent-child relationships are among the most important factors. A meta-analysis conducted by Baude et al. (2016)

that examined 19 studies revealed better child adjustment in SPC compared to sole care arrangements, even though effect sizes were quite low. Moreover, differences in child adjustment were moderated by the time spent with both parents, so that clear SPC-related advantages for child development only emerged if the time between the parental residences was shared equally (i.e., 50:50 time split).

In summary, it can be noted that the study evidence is mixed, depending on which contexts are used, how SPC is defined, and which outcomes are considered for the children. Altogether the studies point to—if any—only minor advantages of SPC compared to sole care models (Bergström et al. 2015; Fransson et al. 2018). Compared to two-parent families there are hardly any disadvantages in child well-being (e.g., Hagquist 2016). Furthermore, the reported associations often disappeared when other family characteristics, such as relationship qualities, were taken into account (Steinbach and Augustijn 2021). In other representative studies, there were hardly any differences between the different physical custody arrangements after separation (i.e., SPC vs. sole care models; Hjern et al. 2021a).

*1.4. SPC and Child Adjustment in the Case of a Conflictual Parental Relationship*

As one of the first, Fehlberg and colleagues (2011) emphasized in their analyses of international and especially Australian data on SPC that SPC may not be the more favorable physical custody arrangement in all cases because it can also present risks under some conditions. For instance, one highly relevant adverse condition, in addition to parents' overall concerns for the safety of their (very young) children, is an ongoing and intense conflict between separated parents that can be exacerbated by SPC. A longitudinal study of McIntosh and colleagues (2010) on families with conflictual separation showed that SPC children had more problems with regard to their attention span and concentration, as well as more problems with task completion than children in the sole care model. This affected boys in particular and especially those who were cared for in a rigidly established SPC model (McIntosh et al. 2010). In this study, SPC children also reported more parental conflict, felt trapped in loyalty conflicts, and were the least satisfied with their custody arrangement after four years. A cross-sectional, large-scale, nationally representative Dutch study revealed similar findings (Kalmijn 2016): conflicts between the separated parents were associated with higher levels of depressive symptoms among children (particularly boys) when there was frequent contact with the father. Similarly, a German cross-sectional study showed that high levels of interparental conflict (Augustijn 2021b), or children's entrapment in loyalty conflicts (Augustijn 2021c), was related to similarly high problem scores on the SDQ among SPC children and those in the sole care model. Positive associations between SPC and child well-being were only found when parents displayed low levels of parental conflict (Augustijn 2021b) or if there were no loyalty conflicts (Augustijn 2021c). A review of 11 studies also indicated that SPC was associated with poor child development in highly conflictual families (Mahrer et al. 2018). This indicates that parental conflicts after a separation (Amato 1993) are even more critical for children in SPC arrangements because children in this arrangement may be exposed to interparental conflicts more often and may feel torn between their parents. Another review of 16 cross-sectional studies concluded that parental conflict and practices, as well as parents' health, were more significant factors for child behavior than the choice of a post-separation custody arrangement (Baude et al. 2019). In contrast, Bauserman's (2002) meta-analysis showed that SPC children experienced fewer conflicts than children in sole care models, but this could not explain differences in the well-being between these children. A recent cross-sectional Swedish study on 12,845 children aged three years showed that the differences in child adjustment (again measured with the SDQ) between post-separation physical custody arrangements disappeared when the parental coparenting quality was taken into account (Bergström et al. 2021). The authors therefore conclude that the parental coparenting relationship plays a more salient role in two-parent and SPC families compared to families practicing sole care models because there is a heightened need for coordination and communication in these models, which requires better cooperation between the parents. Because studies on the association between child

adjustment and coparenting problems in SPC families are predominantly based on cross-sectional data, no conclusions can be drawn about whether child adjustment is predictive of parents' choice to practice SPC, whether SPC leads to more parental coparenting problems, or whether coparenting problems in SPC families lead to adverse child adjustment.

Against this backdrop, the present study focuses on the following research questions:

**RQ1:** Do we observe selectivity into post-separation custody arrangements in Germany?

**Hypothesis 1a.** *More specifically, we expect that the age of the child, the number of children in the household, characteristics of the mothers (i.e., level of education and employment status), and the distance between the parental residences are relevant factors for parents' choice of SPC.* The number of children in the household is particularly interesting because, to our knowledge, prior studies rarely considered this family-level factor. Yet it is plausible to assume that the presence of multiple children in the household may reduce the likelihood to practice SPC because it may be too complex logistically and not feasible to arrange and navigate (potentially different or competing) SPC schedules for multiple children.

**Hypothesis 1b.** *Furthermore, we expect that a more cooperative coparenting relationship between the parents is linked to a higher likelihood to practice SPC.* Given the lack of legal regulations for SPC in Germany, we assume that less conflictual parents are more likely to opt for SPC.

**RQ2:** Do we observe differences in child adjustment between different post-separation custody arrangements in Germany?

**Hypothesis 2a.** *On the one hand, we assume that SPC children display slightly higher levels of child adjustment than children in sole care models.*

**Hypothesis 2b.** *On the other hand, we assume that the quality of the parental relationship has a moderating effect on the link between custody arrangement and child adjustment.* Previous studies have shown that parental conflicts are harmful and can lead to more problems for SPC children, or at least cancel out the advantages of SPC. We argue that a similar mechanism is at play for the parental coparenting relationship, which has rarely been considered in previous research so far. Because coparenting, and in particular conflictual coparenting behaviors among parents (e.g., to "stab one another in the back"), has been shown to be more consequential for child development compared to interparental conflicts in general (Teubert and Pinquart 2010), our study put a special emphasis on this issue.

## 2. Materials and Methods

We used data from the third large-scale, representative German survey of children, youth, and young adults ("Growing up in Germany"; Kuger et al. 2021), which was collected via personal standardized computer-assisted interviews in 2019. The sample includes target persons aged 0 to 32 years randomly drawn from residents' population registers. Once target persons—or the primary caretakers for minors—were willing to participate in the study, the other household members were also recruited to take part in the study (e.g., parents and siblings). Each participant completed a modularized interview covering a wide set of age-graded topics, such as the socio-economic conditions of the household, family life and relationship quality within the family, stressors and strains of growing up, indicators of well-being of all household members, as well as childcare, schooling, occupational training, and work. Those who took part in the study received a small compensation. The response rate was about 21% of households, with target persons initially drawn from the population registers. The full sample comprised 14,277 interviews with target persons and 6621 parent interviews for minors, both nested in 6355 households.

For the purpose of our study, we limited our analytical sample to the subgroup of minors (aged 0 to 17 years) with separated or divorced parents and information on their post-separation physical custody arrangement, as well as the coparenting relationship between their parents. The resulting analytical sample, which we then used to answer our first research question on the prevalence and sociodemographic composition of families in different physical custody arrangements, consisted of 800 minors nested in 509 households (46.6% female; M (SD)$_{age}$ = 9.97 (4.60)). Addressing our second set of hypotheses on the link between physical custody arrangements, parents' coparenting relationship, and child adjustment, our analytic sample was smaller because our main outcome variable (child adjustment) was only assessed for children aged 4 to 17 years and only for one child within each household. Thus, the resulting analytical sample was comprised of 467 children (47.3% female; M (SD)$_{age}$ = 11.74 (4.02)).

### 2.1. Measures

We differentiated four *post-separation physical care arrangements* based on the amount of contact between the child and parent who was not considered to be the primary caretaker (i.e., usually the father) and the monthly count of children's overnight stays at each parental home. Note that father-child contact could comprise both face-to-face contact or other remote forms (e.g., [video] calls or texts) and its amount was measured with a categorical indicator (1 = never, 2 = less than once a month, 3 = once or twice a month, 4 = once or twice a week, 5 = multiple times per week, 6 = daily). Families practiced (1) *SPC* if the child spent at least 10 (30%) and up to 21 nights (70%) per month at the other parental home; (2) *sole care with frequent father-child-contact* if the child spent less than 10 nights per month at the other parental residence and father-child contact was at least once or twice a week or more often; (3) *sole care with occasional father-child-contact* if the child spent less than 10 nights per month at the other parental residence and father-child contact was at least once or twice a month or less often; and (4) *sole care with no contact to the father* if there were no overnight stays at the other parental residence and no father-child contact.

In our full analytical sample (i.e., before the drop in the sample size because of the more restricted child adjustment measure), 8.5% of the children with separated parents practiced SPC, 50.9% lived predominantly with their mother and had frequent contact with the non-residential father, 33.0% of the children had occasional contact with the non-residential father while living predominantly with their mother, and 7.6% of the children with separated parents lived with their mother and had no contact with their non-residential father. Note that the distribution of physical care arrangements in our sample does differ slightly from previous findings reported for Germany (Walper et al. 2021); this is mainly the case because our analytic sample was restricted to families where parents reported to have at least a somewhat active coparenting relationship with the other parent. Because this requires a certain amount of communication between the separated parents, cases of children who had no contact with their father were underrepresented, whereas more frequent contact with the father and SPC were reported more often. If we look at the distribution of post-separation care arrangements for all minors with separated parents in our dataset (i.e., those with and without information on parents' coparenting relationship; *N* = 1165), 6.1% of children with separated parents practiced SPC, 39.5% had frequent contact with the non-residential father while predominantly living with their mother, 29.7% lived predominantly with their mothers and had occasional contact with their non-residential father, and 24.7% of the children with separated parents had no contact with their non-residential father.

The *distance between the parental residences* is an important factor in determining the choice of post-separation care arrangements (Walper et al. 2021; Schier and Hubert 2015; Kaspiew et al. 2009). In our study, respondents were asked to rate whether the parental residences were: 1 = in the same house, 2 = in the same neighborhood, 3 = in the same town or village, but more than 15 min away, 4 = in a different town or village, but less than one hour away, 5 = further away, but in Germany, or 6 = further away, in another country. We collapsed the first two categories (i.e., in the same house and neighborhood) because of the low case number for the first category and then entered this indicator as a continuous variable (ranging from 1 "same house/neighborhood" to 5 "further away, in another country) into our models. This approach is common practice and recommended for ordinal indicators with at least five categories (e.g., Rhemtulla et al. 2012; Robitzsch 2020).

The *quality of parental coparenting* assesses how well ex-partners work together on parenting issues. Because coparenting requires at least a minimum amount of contact and exchange between the ex-couple, questions on this measurement were restricted to mothers who had contact with the biological father. We used an adapted version from the German translation of the Parent Problem Checklist to measure the quality of the coparenting relationship (Dadds and Powell 1991). Six items (Cronbach's alpha = 0.80), namely "We are a good team as parents", "We make important decisions concerning our child together", "We have fundamentally different ideas about parenting", "We stab each other in the back", "He/she drags our child into our conflicts", and "Discussions about parenting practices often end with us fighting", were rated on a scale from 1 "completely disagree" to 6 "completely agree". Note that for our analyses, a composite score was formed and items were recoded in a way that higher values correspond to higher levels of coparenting problems between ex-partners.

We used the *Strengths and Difficulties Questionnaire* SDQ; Goodman 2006 to measure children's psychosocial adjustment. This broadly validated instrument is well-suited to examine child and youth problem behavior for minors aged 4–17 years. The SDQ consists of five subdimensions (with five items each), namely *emotional problems*, *conduct problems*, *hyperactivity*, *peer problems*, and *prosocial behavior* (0 = not true, 1 = somewhat true; 2 = certainly true). As mentioned before, the SDQ was filled out by mothers for only one child aged 4 to 17 years in the household (if there were multiple children in the household within this age range) in our study. In most cases, this child was the oldest in the household. Again, higher values on the subscales indicate worse child adjustment.

Mothers further provided detailed information about the household's socio-demographic composition, its socio-economic situation, and their own educational attainment. We have information on *maternal employment status* (1 = not employed [e.g., because of unemployment, post-secondary schooling or vocational training, or maternity leave; 2 = marginal or part-time employment; 3 = full-time employment). *Mother's educational attainment level* was classified according to the Comparative Analysis of Social Mobility in Industrial Nations (Brauns et al. 2003), which takes both the level of schooling and the post-secondary level of occupational or academic training into account. Due to small case numbers, we collapsed some categories, namely 1 = low, if mothers had a high school diploma only; 2 = medium, for vocational training; and 3 = high, for some college or more. In our sample, 39.4% of mothers were highly educated, 39.6% had medium levels of education, and 20.9% had low levels of education In order to assess the financial situation of the household, we also measured *families' level of perceived economic deprivation*. Mothers rated the following statements with regard to their financial situation: "We can put away money each month," "We are able to replace furniture," and "We are able to pay for unexpected expenses" (1 = yes; 2 = no because of financial reasons; 3 = no because of other reasons). Negative answers were counted and a three-level categorical indicator was formed (1 = none; 2 = low [i.e., count of one]; 3 = high [i.e., count of two or three]). Finally, we have information on the *age and* (in full years) and *sex of the child* (0 = female, 1 = male), as well as the *number of children living in the maternal household* (1 = one child; 2 = two children; 3 = three or more children).

*2.2. Analytic Strategy*

All analyses were conducted in Stata (v15.1). To address the first research question on the selectivity into SPC, we ran multinomial logistic regression models predicting the likelihood of belonging to each post-separation custody arrangement. Because multiple children can be nested within one household, we used cluster-robust standard errors in our models. The reference category was SPC. In a first step, we added the distance between the parental residences and the socio-demographic, as well as socio-economic indicators (e.g., mother's educational attainment and employment status, as well as families' level of perceived economic deprivation and children's age and sex) into the model (Hypothesis 1a).

*In a second step, the quality of the coparenting relationship was added to the model* (Hypothesis 1b). To ease interpretation, we report discrete differences in average marginal effects (AME) of the regression models (Long 2014). AME represent the average impact of the independent variable on the likelihood of each outcome category (i.e., belonging to each post-separation care arrangement). For continuous variables, AME represents the average discrete change in the predicted probabilities for a one-unit increase in the predictor and, for categorical variables it expresses average differences in predicted probabilities for pairs of levels of the predictor.

In order to answer our second research question on the link between care arrangements, parental coparenting, and child-wellbeing, we used linear regressions models predicting child adjustment problems (i.e., the four SDQ subscales). Again, we proceeded in two steps. First, we ran a main effect model including all predictors (e.g., post-separation care arrangements, the socio-demographic and socio-economic indicators, as well as the quality of the parental coparenting relationship (Hypothesis 2a).

In a second step (Hypothesis 2b), we added interaction terms between post-separation care arrangements and coparenting to test whether children in SPC arrangements display higher levels of child adjustment depending on the quality of the parental coparenting relationship. To ease the interpretation of significant interaction terms, we estimated and plotted predictive margins.

### 3. Results

*3.1. Multinomial Regression Results*

The results testing our first hypotheses on the selectivity into physical custody arrangements are presented in Figures 1 and 2.

Results of the first multinomial regression model show the predicted probabilities of belonging to each respective post-separation care arrangement (see Figure 1). It can be seen that only a few significant differences emerged across the different physical custody arrangements. However, the distance between the two parental residences was an important factor for all four post-separation physical custody arrangements. If the ex-partners lived further away from each other, children were more likely to reside with their mothers and to have either no (AME (SE) = 0.03 (0.01), $p < 0.01$) or only occasional contact with their nonresidential father (AME (SE) = 0.08 (0.02), $p < 0.001$). In contrast, if the two parental residences were in closer proximity to each other, the chance that children lived primarily with their mothers and had frequent father-child-contact (AME (SE) = $-0.05$ (0.02), $p < 0.05$), or practiced SPC (AME (SE) = $-0.05$ (0.01), $p < 0.001$), was higher. Note that we reran our models entering the distance between the parental residence as a categorical indicator as an additional sensitivity check (reference category is same house/neighborhood; results available upon request). Results revealed that there was no significant difference in children's likelihood to practice SPC when the parental residences were in the same house/neighborhood vs. in the same town/village, but 15 min away. Children were more likely to be in SPC, however, when the parental residences were in the same house or neighborhood compared to those who reside in a different town/village, elsewhere in Germany, or abroad (both less than and more than an hour away). Relatedly, children were more likely to reside with their mothers and to have no father-child contact if the parental

residences were in a different town/village, elsewhere in Germany, or abroad (both less and more than one hour away) compared to within the same house or neighborhood.

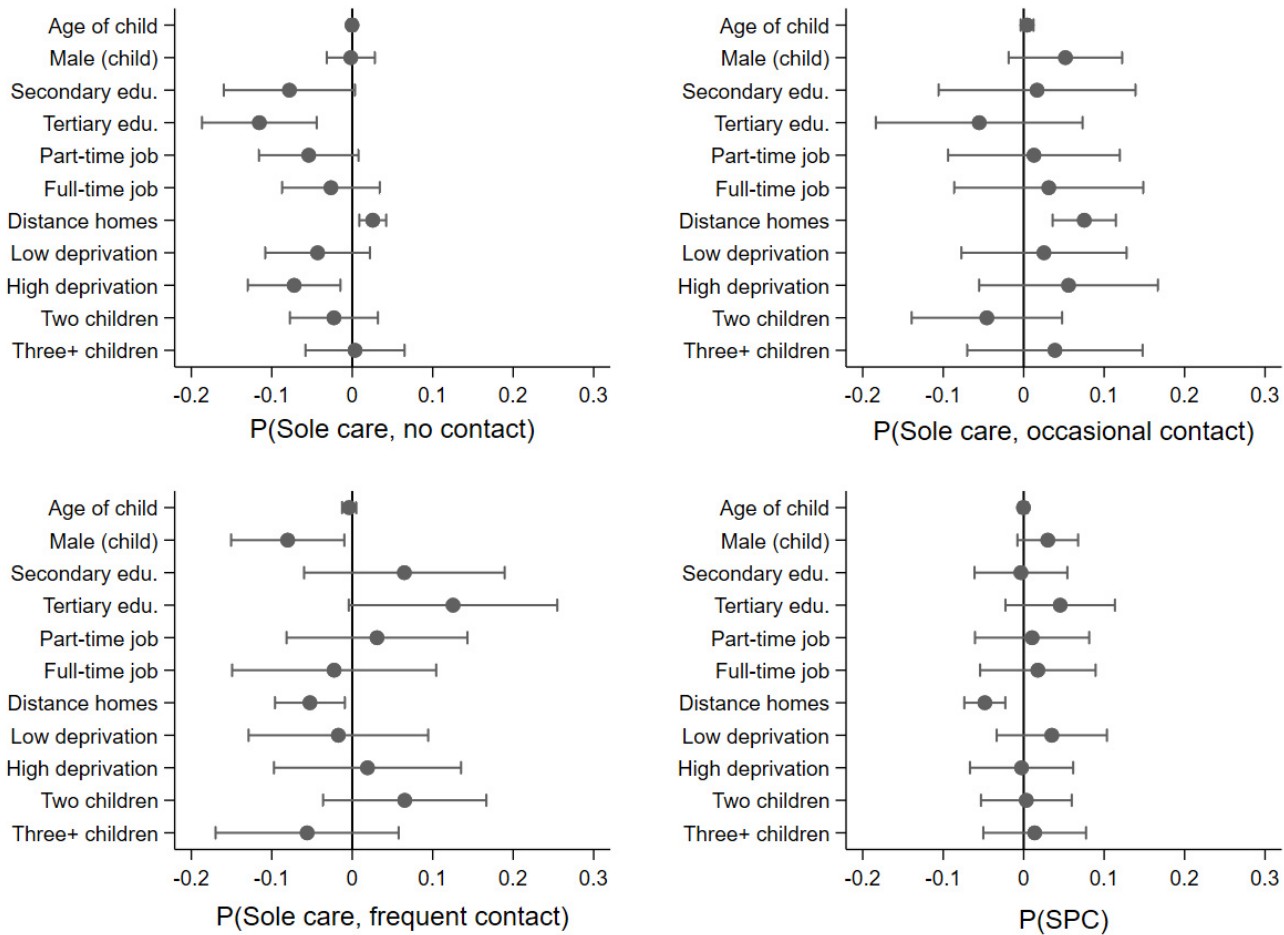

**Figure 1.** Predicted probabilities of belonging to each post-separation physical custody arrangement based on socio-demographic and socio-economic predictors. Reference categories are: Female; Primary levels of education; Not employed; No economic deprivation; One child in the household.

In our main set of analyses displayed in Figure 1, children with mothers who had tertiary levels of education were further less likely to belong to the sole care with no father-child contact group compared to children where mothers had primary levels of schooling (AME (SE) = −0.12 (0.04), $p < 0.01$). However, children were also more likely to belong to the sole care with no father-child contact group if mothers were less likely to report experiencing high levels compared to no economic deprivation (AME (SE) = −0.07 (0.03), $p < 0.05$). This is somewhat unexpected because previous research has documented the strong link between single motherhood and the increased risk of poverty (e.g., Chzhen and Bradshaw 2012; Heintz-Martin et al. 2021). Lastly, boys were less likely to live with the mother and to have frequent father contact compared to girls (AME (SE) = −0.08 (0.04), $p < 0.05$), which is in line with the finding by Kalmijn (2016) that boys were more likely to live in SPC families. This tendency can also be seen here, but is not significant for SPC.

When we entered the quality of parental coparenting into our model (see Figure 2), results for the predictors already entered in Step 1 remained largely unchanged. However, the distance between the two parental residences was no longer significant for children in sole care arrangements with either no or frequent father-child contact. Boys were also no longer less likely to live with their mothers and to have frequent father contact compared to girls. Yet children, where mothers had either secondary or tertiary levels of education, were less likely to belong to the sole care with no father-child contact group compared to children

where mothers had primary levels of schooling (AME (SE) = −0.08 (0.04), $p < 0.05$, and AME (SE) = −0.11 (0.03), $p < 0.01$, respectively). The quality of parents' coparenting relationship was further strongly associated with all four post-separation care arrangements. Children were more likely to either be in SPC or to reside with their mothers, but to have with frequent father-child contact, if their parents had a less conflictual coparenting relationship (AME (SE) = −0.03 (0.01, $p < 0.01$, and AME (SE) = −0.08 (0.02), $p < 0.001$, respectively). In contrast, children were more likely to live with their mothers and to either have no or only occasional father-child contact if their parents had a more coparenting problems (AME (SE) = 0.03 (0.01), $p < 0.001$, and AME (SE) = 0.08 (0.01), $p < 0.001$, respectively).

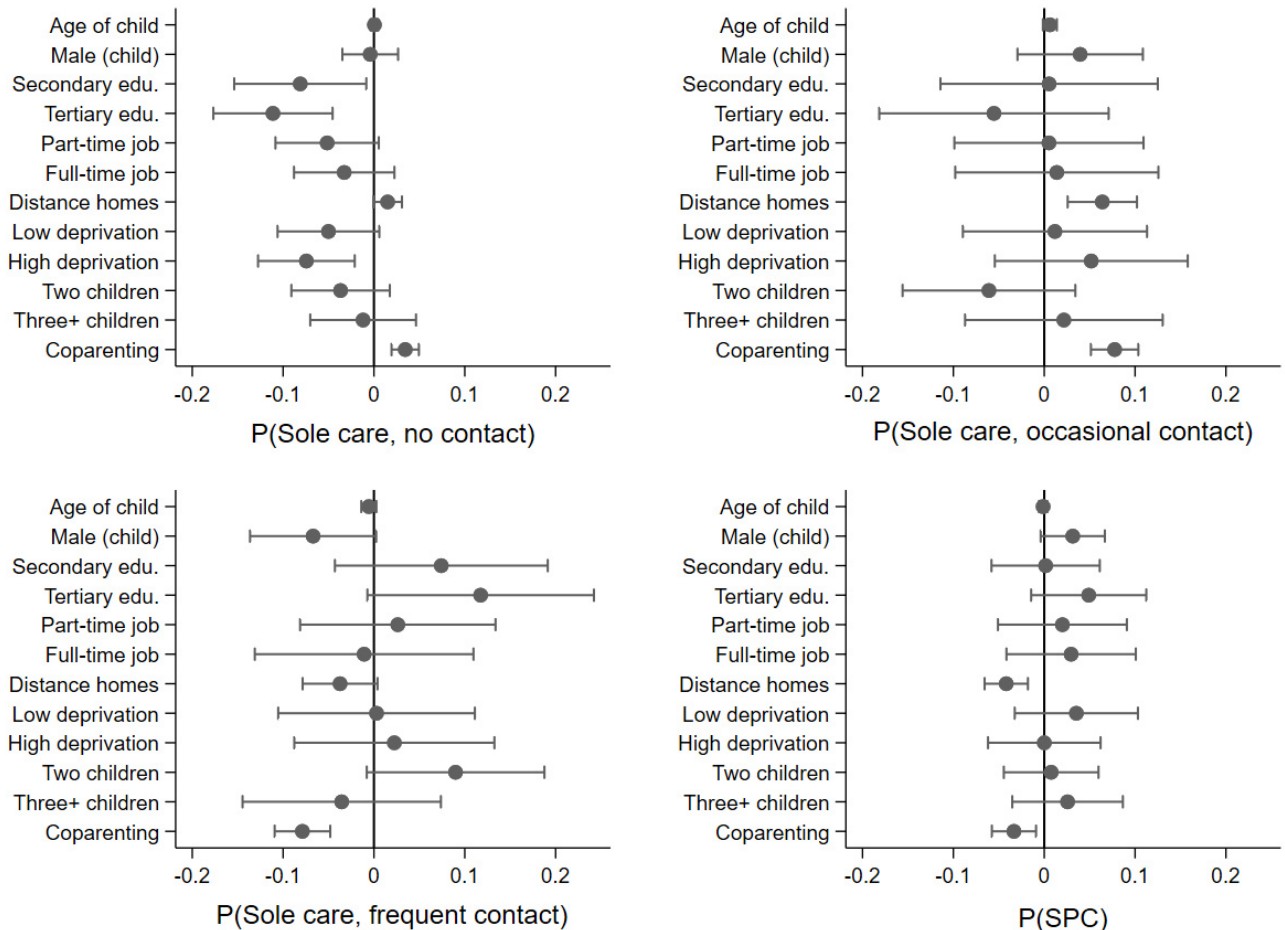

**Figure 2.** Predicted probabilities of belonging to each post-separation physical custody arrangement based on socio-demographic and socio-economic predictors, as well as the quality of parental coparenting relationship. Reference categories are: Female; Primary levels of education; Not employed; No economic deprivation; One child in the household.

### 3.2. Linear Regression Results

To examine our second set of hypotheses (2a and 2b), namely on the links between physical custody arrangements, the quality of the parental coparenting relationship, and child adjustment, we used stepwise OLS regression models, as presented in Table 1. For each of the four subdimensions of the SDQ (emotional problems, conduct problems, hyperactivity, and peer problems), we first ran a main effect model (Model 1) that included the physical custody arrangements, the quality of the parental coparenting relationship, as well as our main socio-demographic and socio-economic predictors. It can be seen that none of the physical custody arrangements were significantly linked to the subdimensions of the SDQ, which was contrary to our expectation. More coparenting problems, however, were significantly linked to higher levels of children's emotional, conduct, and peer prob-

lems, which is in line with previous studies (Langmeyer 2015; Teubert and Pinquart 2010). Furthermore, we found differences in children's psychosocial adjustment by age, gender, and levels of maternal education, which is also well-documented in previous research (e.g., Woerner et al. 2004). For instance, mothers of older children reported significantly fewer emotional and conduct problems, as well as lower levels of hyperactivity. Mothers also reported significantly more conduct problems and higher levels of hyperactivity for boys compared to girls. For three out of four SDQ subscales (namely emotional, conduct, and peer problems), higher levels of maternal education were associated with fewer problems on these subscales. For conduct problems only, we found that mothers who were employed full-time reported that their children displayed more conduct problems compared to mothers who were not employed. Lastly, mothers who report high levels of economic deprivation were more likely to report children's peer problems compared to those with no economic deprivation.

In a second step, we added the interactions between physical custody arrangement and the quality of the parental coparenting relationship into the model (see Table 1, Model 2) to test whether children in SPC may display lower levels of child adjustment depending on the quality of the parental coparenting relationship (Hypothesis 2b). There was only one significant interaction between physical custody arrangements and coparenting problems for the subdimensions hyperactivity, but not for the other three subdimensions emotional, conduct, and peer problems. Predictive margins plotted in Figure 3 showed that, among children in SPC, a more conflictual coparenting relationship between the ex-partners was associated with higher levels of hyperactivity. In contrast, a more conflictual coparenting relationship was associated with lower levels of hyperactivity among children who lived with their mothers and had no contact with their fathers.

**Table 1.** Stepwise OLS regression models predicting the SDQ subscales (one model each; higher values indicate worse child adjustment; Model 1 = main effect model; Model 2 = with interactions between post-separation physical custody arrangements and the quality of the parental coparenting relationship, $N = 434$).

| Predictor | Emotional Problems | | Conduct Problems | | Hyperactivity | | Peer Problems | |
|---|---|---|---|---|---|---|---|---|
| | Model 1 | Model 2 | Model 1 | Model 2 | Model 1 | Model 2 | Model 1 | Model 2 |
| Sole care/no contact [a] | 0.10 (0.54) | 1.11 (1.64) | 0.13 (0.40) | 0.96 (1.22) | 0.65 (0.62) | 4.01 * (1.87) | 0.03 (0.42) | 1.50 (1.29) |
| Sole care/occasional contact [a] | −0.17 (0.39) | −1.40 (0.90) | −0.09 (0.29) | −0.53 (0.67) | −0.18 (0.45) | 1.58 (1.03) | 0.03 (0.31) | 0.23 (0.71) |
| Sole care/frequent contact [a] | −0.17 (0.36) | −0.20 (0.81) | −0.24 (0.27) | 0.09 (0.60) | −0.26 (0.42) | 1.92 * (0.92) | −0.08 (0.28) | −0.02 (0.63) |
| Age of child (in full years) | −0.05 * (0.02) | −0.05 (0.03) | −0.08 *** (0.02) | −0.08 *** (0.02) | −0.18 *** (0.03) | −0.17 *** (0.03) | −0.01 (0.02) | −0.00 (0.01) |
| Male | 0.22 (0.19) | 0.21 (0.19) | 0.37 * (0.14) | 0.36 * (0.14) | 0.75 ** (0.22) | 0.76 ** (0.22) | 0.49 ** (0.15) | 0.49 ** (0.15) |
| Secondary education [b] | −0.16 (0.27) | −0.12 (0.27) | −0.32 (0.20) | −0.31 (0.20) | 0.19 (0.31) | 0.14 (0.31) | 0.05 (0.21) | 0.07 (0.21) |
| Tertiary education [b] | −0.56 * (0.28) | −0.50 (0.28) | −0.58 ** (0.21) | −0.54 * (0.21) | −0.18 (0.32) | −0.12 (0.32) | −0.22 ** (0.22) | −0.20 (0.22) |
| Part-time employed [c] | 0.25 (0.25) | 0.26 (0.25) | 0.09 (0.19) | 0.09 (0.19) | 0.05 (0.29) | 0.01 (0.29) | 0.12 (0.20) | 0.11 (0.20) |
| Full-time employed [c] | 0.31 (0.28) | 0.30 (0.28) | 0.44 * (0.21) | 0.44 * (0.21) | 0.23 (0.32) | 0.22 (0.32) | 0.27 (0.22) | 0.26 (0.22) |
| Distance between homes | −0.11 (0.10) | −0.09 (0.10) | −0.06 (0.07) | −0.05 (0.07) | −0.05 (0.11) | −0.06 (0.11) | −0.06 (0.08) | −0.05 (0.08) |
| Low deprivation [d] | −0.13 (0.24) | −0.14 (0.24) | −0.16 (0.18) | −0.17 (0.18) | −0.18 (0.28) | −0.19 (0.28) | 0.17 (0.19) | 0.17 (0.19) |
| High deprivation [d] | 0.43 (0.26) | 0.37 (0.26) | 0.24 (0.19) | 0.20 (0.19) | 0.41 (0.29) | 0.37 (0.29) | 0.59 (0.20) | 0.58 ** (0.20) |

**Table 1.** *Cont.*

| Predictor | Emotional Problems | | Conduct Problems | | Hyperactivity | | Peer Problems | |
|---|---|---|---|---|---|---|---|---|
| | Model 1 | Model 2 | Model 1 | Model 2 | Model 1 | Model 2 | Model 1 | Model 2 |
| Two children [e] | 0.30 (0.23) | 0.26 (0.23) | 0.33 (0.17) | 0.30 (0.17) | 0.49 (0.26) | 0.51 (0.26) | 0.09 (0.18) | 0.10 (0.18) |
| Three and more children [e] | 0.38 (0.26) | 0.34 (0.26) | 0.37 (0.20) | 0.34 (0.20) | 0.24 (0.30) | 0.20 (0.30) | 0.12 (0.21) | 0.11 (0.21) |
| Coparenting problems | 0.20 * (0.08) | 0.08 (0.27) | 0.20 ** (0.06) | 0.24 (0.20) | 0.07 (0.09) | 0.85 ** (0.31) | 0.14 * (0.06) | 0.20 (0.21) |
| Sole care/no contact × Coparenting [f] | | −0.22 (0.46) | | −0.23 (0.34) | | −1.16 * (0.52) | | −0.41 (0.36) |
| Sole care/occasional contact × Coparenting [f] | | 0.38 (0.30) | | 0.11 (0.22) | | −0.73 * (0.34) | | −0.08 (0.23) |
| Sole care/frequent contact × Coparenting [f] | | 0.03 (0.29) | | −0.12 (0.21) | | −0.88 ** (0.33) | | −0.03 (0.23) |
| Ajdusted R² | 0.04 | 0.04 | 0.11 | 0.11 | 0.11 | 0.12 | 0.04 | 0.04 |

Notes: Reference categories are: [a] SPC; [b] primary levels of education; [c] not employed; [d] no economic deprivation; [e] one child in the household, [f] SPC × Coparenting. Cells represent B (SE). * $p < 0.05$. ** $p < 0.01$. *** $p < 0.001$.

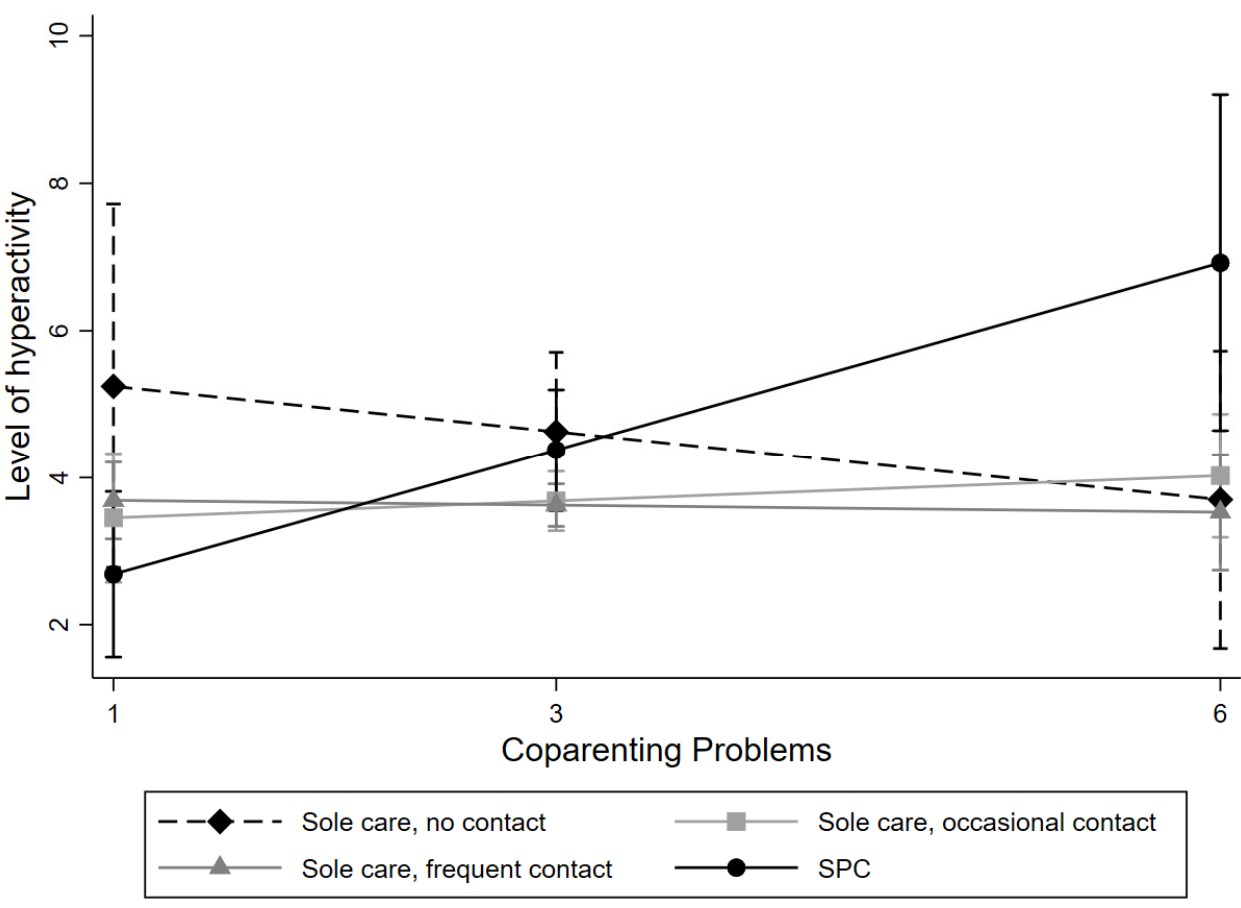

**Figure 3.** Significant interaction effect between post-separation physical custody arrangement and coparenting problems for the SDQ subdimension hyperactivity.

## 4. Discussion

This study contributes to the emerging literature on SPC in jurisdictional contexts lacking legal and institutional support for SPC in contrast to many other Western nations (e.g., the Nordic countries, parts of the U.S., and Australia; Steinbach 2019). We examined

the prevalence of SPC in Germany, issues related to social selection into SPC, and its associations with child adjustment. More specifically, we compared children in SPC arrangements to their peers in the so-far largely predominant sole care model, where children tend to reside with their mothers and father-child contact is often limited (Swiss and Bourdais 2009; Köppen et al. 2018). Families practicing SPC tend to be more affluent, better educated, less conflictual, and to live in closer proximity (Nielsen 2013). Both parents further have the chance to remain engaged in childcare-related tasks and duties when practicing SPC. Some scholars have argued that this could lead to better child adjustment, yet findings on linkages between SPC and child adjustment remain mixed (Baude et al. 2016; Bergström et al. 2015), especially under the condition of a difficult, conflictual parental relationship. To address these inconsistent results, our study profiled the German case for which only a few studies on the prevalence of and associations with SPC exist to date (e.g., Augustijn 2021b; Köppen et al. 2020; Walper et al. 2021). Our first aim was to provide an estimate of the share of children of separated parents whose parents practice SPC based on very recent representative survey data collected in 2019 and to examine the sociodemographic profile of SPC parents compared to those who adopt more traditional sole care models. In this context, the differentiated view of the sole care models is an added value of the present study, as these have often been compared as one group to SPC in previous studies (e.g., Augustijn 2021a). We found that, in concordance with other German studies (Köppen et al. 2020; Walper et al. 2021), only a small fraction of children in post-separation families in our sample live in SPC (about 6%) compared to shares of more than 30% of children of separated parents living in SPC in other European countries (e.g., Belgium, Sweden, or Spain; Vanassche et al. 2017). This indicates that choosing SPC still represents an exception rather than the rule in Germany and that more traditional arrangements of sole maternal care are predominantly practiced by the vast majority of separating parents. It should be noted that children were the unit of analysis in our study, whereas other studies often conducted their analyses on the family level. Readers should keep this in mind when comparing estimates across studies, such as for the prevalence of SPC. The dominance of sole care arrangements can, in part, be explained by the lack of institutional and legal support for SPC in Germany to date, as well as with the likely prevailing gendered role distributions among German couples before and after a separation or divorce (e.g., Grunow et al. 2018). This means that SPC parents have to negotiate and agree on SPC in large parts on their own, which requires a substantial amount of agency, whereas the existing set of institutional regulations are more likely to channel and filter most other parents into sole care arrangements (Mills and Blossfeld 2009; Settersten and Gannon 2005). The lack of supportive family policies, such as regional shortcomings in the provision of public childcare that pose a considerable barrier for maternal employment (Boll and Lagemann 2019), further reinforce a gendered division of post-separation custody arrangements (Levy 2013). However, Fabricius and colleagues (2010) found that if divorce professionals and counselors held more favorable views of SPC and guided post-separation families accordingly, rates of SPC tended to change substantially even in the absence of legal support for SPC. This could therefore be a promising avenue to support parents in their choice to practice SPC after a separation or divorce in light of the lack of legal support for SPC in Germany. We further examined sociodemographic factors related to parents' choice to opt for SPC rather than for other sole care custody arrangements where children reside with their mothers and vary in the degree of contact with the nonresidential father (i.e., ranging from no to frequent father-child contact). Overall, our results were in line with prior studies indicating that the probability of practicing SPC is strongly associated with the proximity between the parental residences (e.g., Schier and Hubert 2015; Walper et al. 2021). Separated families were more likely to practice SPC, or have at least frequent contact when practicing a sole care model if the parental residences were in closer geographical proximity. If the father lived relatively far away from his child, the chance to have little or no contact was significantly higher. This is likely related to the feasibility and costs associated with different post-separation custody arrangements. Practicing SPC may simply not be feasible and possible on a regular

basis, as well as represent a rather costly endeavor if children need to travel long distances between the parental homes (Schier and Hubert 2015). However, parents' selection of sole care models rather than SPC, and a greater geographic distance between parental residences, could also all be the result of a father lacking interest in child involvement or lacking emotional closeness to children, as well as due to severe conflicts between the parents. These alternative explanations need to be kept in mind, though we were not able to test them because of data availability issues.

Another well-known factor associated with SPC is parental SES and particularly parents' educational attainment (Fehlberg et al. 2011; Juby et al. 2005; Poortman and van Gaalen 2017). In our data, we also found a significant link between maternal education and post-separation custody arrangements, although only for sole care arrangements where children had no contact with the nonresident father. Mothers with tertiary levels of schooling were less likely to belong to this post-separation custody arrangement compared to mothers with primary levels of schooling, which is in line with prior studies on the precarious and more disadvantaged economic situation of single mothers specifically (Bernardi et al. 2018; Chzhen and Bradshaw 2012). In this context, the somewhat unexpected finding that high levels of economic deprivation decrease the chance to practice sole care where children had no contact with the nonresident father compared to those with no economic deprivation could be related to regulations concerning child support in Germany. Previous research has shown that nonresidential fathers who have no contact with their children often also provide little or no financial child support (Juby et al. 2007; Keil and Langmeyer 2020). In this case in Germany, the state steps in and provides at least parts of the fathers' financial obligations to the mother. This could mean that precisely mothers of these children with no father-child contact receive more frequent and regular financial support from the state, compared to mothers where children have sparse or frequent contact with the nonresidential father, where fathers often pay too little support and only irregularly (Hubert et al. 2020). If there is any kind of contact with the nonresident father, the state naturally has an interest in demanding child support payments directly from the father, which probably is why we did not observe this effect for the other sole care groups. Contrary to our hypotheses and the previous literature (Walper et al. 2021), we do not find any significant correlations of the age of the child as well as the number of children with the post-separation physical custody arrangements.

Lastly, we also considered the quality of the parental coparenting relationship as a factor shaping parents' choice and selection into different post-separation custody arrangements (Walper et al. 2021; Cashmore et al. 2010; Poortman and van Gaalen 2017) and child adjustment (Beckmeyer et al. 2014; Lamela et al. 2016; Amato et al. 2011), as well as an effect modifier of the relationship between post-separation custody arrangement and child adjustment. The importance of parental coparenting for child adjustment, the choice of post-separation custody arrangements, and its ripple effects can, in part, be explained with the linked lives principle of the life course perspective (Settersten 2015; Elder and Shanahan 2006), which highlights the interconnectedness and social influence among family members across the life course (Thomas et al. 2017). More specifically, individuals are and continue to be embedded in an interwoven and complex web of family ties consisting of the ex-couple, their children, and potentially new partners and biological or stepchildren after a separation or divorce, which fulfill different emotional, social, and material needs. Tensions in parents' ability to work together as a team with regard to their parenting duties and responsibilities can therefore affect children's adjustment and well-being directly and comprehensively (e.g., Walper et al. 2021), as well as through spillover processes on the mothers' use of harsher parenting practices (e.g., Heintz-Martin et al. 2021; Choi and Becher 2018). Because SPC requires more communication and exchanges between the parents, less conflictual parents are more likely to practice SPC (Poortman and van Gaalen 2017), which is what we also saw in our analyses. We found that mothers who reported to have more coparenting problems with the other parent were more likely to practice sole care where children have little or no contact with the nonresidential father, whereas a less conflictual coparenting

relationship was associated with a higher chance to practice SPC or sole care models where children have frequent contact with the nonresidential father. These results are further in line with seminal work on the consequences of parental conflict and divorce on child adjustment showing that children tend to fare better in a less conflictual single-parent family rather than in a conflict-ridden two-parent family (e.g., Amato 2000; Cherlin et al. 1991; Demo and Acock 1988; Hess and Camara 1979).

However, it is important to note that our sample was restricted to only cases with an active co-parenting relationship for these analyses. It is possible that among—most likely highly-conflictual—post-separation families where no active coparenting relationship exists, levels of child maladjustment and hyperactivity specifically are equally high or even more amplified. Furthermore, whether the parents' more cooperative and trusting relationship with regard to parenting issues after a separation is the cause for or a consequence of the chosen physical custody arrangement and the amount of father-child contact remains an open question because our data is cross-sectional and does not allow us to draw any causal inferences. Because SPC parents tend to be better educated, and less conflictual, SPC children could be expected to be better adjusted compared to those in more traditional sole care arrangements. However, we did not find any significant associations between the different physical custody arrangements and children's psychological adjustment, measured with four different dimensions of the SDQ (Goodman 2006). This is in line with some prior studies reporting no or minor differences in child well-being between children in SPC and children in sole care models (Hjern et al. 2021a; Baude et al. 2016) and does not support findings showing such differences (Augustijn 2021a). The lack of significant differences in child adjustment between SPC and sole care models could be related to the time split we used to define SPC in our study. It could be the case with the rather unequal 70/30% time split, where the majority of care duties and child-related expenses still need to be shouldered by one parent, differences between SPC children and children in the sole care models with frequent-father child contact are rather marginal. Unfortunately, we had to rely on this time split, which is also commonly used in other studies (e.g., Sodermans et al. 2013; Recksiedler and Bernardi 2021), because a stricter definition of 50/50% or 60/40% time split would have resulted in very low case numbers for SPC children and parents. We nevertheless saw that the quality of the parental coparenting was associated with children's emotional, conduct, and peer problems, which we also expected based on the linked lives principle, and that levels of hyperactivity were higher only among SPC children if parents had more coparenting problems. This could be the case because SPC requires constant communication and contact with the ex-partner compared to sole care models with less paternal involvement (e.g., to discuss children's daily schedules or logistics with regard to children's transitions between the parental homes). Consequently, if these exchanges are rather conflictual, they may have particularly adverse effects on children. Coparenting conflicts that relate to the children and that are acted out in front of the children are particularly stressful for children (Teubert and Pinquart 2010). Thus, this could indicate that SPC, where children live in both parental households, may expose children to parental conflicts more and may place children into more intense loyalty conflicts. It could therefore be the case that SPC is only beneficial for child well-being and adjustment if parents are not highly conflictual and resume a constructive and civil parenting relationship. For children in sole care models with no father-child contact, in contrast, a lower coparenting quality between the mother and the non-residential father was associated with lower levels of hyperactivity. In this case, the more conflictual parental coparenting relationship may not be as detrimental for children's psychosocial adjustment precisely because of the limited contact between the parents and between the father and child. These mothers may serve as gatekeepers in particularly conflictual coparenting relationships with the non-residential father by shielding their children intentionally from the adverse effects of the tensions between the ex-couple (Austin et al. 2013). One explanation why this moderation effect is only evident in the case of hyperactivity could be that the other problem domains, such as emotional problems, are not as visible and presently perceived by the mothers.

### 5. Limitations and Conclusions

Our study has several limitations. First, and as mentioned above already, our models are based on cross-sectional data and can therefore only reflect one single snapshot of physical custody arrangements, parental coparenting, and child adjustment after the separation or divorce. This makes it impossible to prove or claim causality and to disentangle complex cause and effect relationships between key constructs in our study. For example, the quality of parents' coparenting relationship is linked to SPC (i.e., is treated as an exogenous variable), but coparenting problems could also arise due to the heightened need for exchanges in SPC arrangements. Furthermore, it could be the case that there was considerable temporal variation in these constructs that is not captured in our data (e.g., changing schedules and routines in the physical custody arrangements or shifting dynamics in the parental coparenting relationship over time). It is therefore crucial that longitudinal studies—ideally prospective studies that begin before a separation or divorce occurs—disentangle the directionality of these effects and address change in the associations between post-separation custody arrangements, parents' coparenting relationship, and child well-being.

Second, our study is limited to mothers' reports on both the quality of the coparenting relationship and child adjustment. It could be the case that fathers' perspectives on both issues differ considerably from mothers' ratings (e.g., Mikelson 2008), which may be particularly relevant for highly conflictual family constellations. Under these circumstances, the information provided by mothers may be more selective or one-sided. Furthermore, prior research has shown that fathers tend to report more problem behaviors with regard to children's level of hyperactivity than mothers, especially if the child was a boy (Chiorri et al. 2015; Davé et al. 2008). Because our study only draws on mothers' perspectives, it is possible that group differences in children's levels of hyperactivity by physical custody arrangements and varying levels of parental coparenting problems could be amplified if father reports were available. Relatedly, it would have been interesting to include more information about non-residential fathers and couples' relationship before union dissolution in our models because paternal involvement before a separation or divorce has been shown to be associated with the choice of post-separation custody arrangements later on (Juby et al. 2005). Unfortunately, no information on fathers' pre-separation involvement in childcare or related issues was available in our dataset. Future studies should examine these pre-separation dynamics as a possible contributing factor to parents' choice of a post-separation physical custody arrangement and examine mothers' and fathers' ratings jointly in dyadic models. Moreover, some of the association between SPC, the co-parenting quality, and child adjustment could be due to method invariance, with all reports coming from a single family member (i.e., the mother). Future studies would therefore benefit from multiple reporters to avoid this methodological problem.

Lastly, in order to draw a more holistic picture of post-separation families, it would also be helpful to have information from the children themselves, especially when links between physical custody arrangements and child adjustment are examined. This is particularly important because ratings on the SDQ provided by parents and children have been shown to differ significantly (Vugteveen et al. 2021). Furthermore, it is also possible that mothers of older children are not necessarily informed about or aware of the amount of contact that their child has with the separated father, especially if the child-father contact occurs on children's mobile devices and online.

Our study provides a comprehensive view on the prevalence and correlates of different post-separation custody arrangements in Germany, where female-headed sole care arrangements are still predominant and SPC is notably infrequent. This can be attributed to the fact that SPC is not yet established by law or encouraged with a substantial amount of institutional support for SPC. The two main conclusions we draw from our findings may therefore only be applicable to similar jurisdictional contexts that lack legal and institutional support for SPC as well. On the one hand, we saw that parents' choice of post-separation physical custody arrangement was still highly selective (e.g., it depends on parents' SES

and especially on mothers' education). This is likely the case because no legal pathways for SPC are put in place in Germany to date, which means that only more affluent and less conflictual parents may be able to afford and implement SPC more successfully. To give all children the opportunity to practice SPC, or to have frequent contact with fathers after a separation, legislation should foster and strengthen these arrangements. Public campaigns for counseling services and divorce professionals targeting post-separation families should also inform parents about the benefits and downsides of SPC as one alternative to more traditional sole care models, which has been shown to change rates of SPC substantially in other contexts (Fabricius et al. 2010). On the other hand, we were able to show that a good and civil relationship between the separated parents, rather than the choice of post-separation physical custody arrangement by itself, is a decisive factor for child well-being. It is therefore imperative that parents are trained and supported to maintain a positive coparenting relationship and to avoid exposing children to ongoing (loyalty) conflicts after a separation (cf. Bergström et al. 2021). This can be fostered through mediation throughout the parental separation or divorce process (Sbarra and Emery 2008). Furthermore, a number of effective coparenting programs (Eira Nunes et al. 2021) already exist for this purpose. These could be implemented and offered more systematically for post-separation families (e.g., through court mandate). It is also essential that these interventions are offered to SPC families if the German family law strengthens parents' access to SPC because coparenting problems have been shown to be particularly harmful to SPC children.

**Author Contributions:** Conceptualization, A.N.L., C.R. and S.W.; Formal analysis, C.R.; Methodology, A.N.L., C.R. and S.W.; Writing—original draft, A.N.L., C.R. and C.E.-P. All authors have read and agreed to the published version of the manuscript.

**Funding:** This research received no external funding.

**Institutional Review Board Statement:** Ethical review and approval was not required for the study on human participants in accordance with the local legislation and institutional requirements. Written informed consent to participate in this study was provided by the participants' legal guardian.

**Informed Consent Statement:** Informed consent was obtained from all subjects involved in the study.

**Data Availability Statement:** Data (https://doi.org/10.17621/aida2019) are publically available for scientific use at: https://surveys.dji.de 13 November 2021.

**Conflicts of Interest:** The authors have no conflict of interest to declare.

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
