# Peer review of "Post-Separation Physical Custody Arrangements in Germany: Examining Sociodemographic Correlates, Parental Coparenting, and Child Adjustment"

_socsci, doi:10.3390/socsci11030114_

Round 1

Reviewer 1 Report

Review of Post-Separation Physical Custody Arrangements in Germany: Examining Sociodemographic Correlates, Parental Coparenting, and Child Adjustment

Journal: Social Sciences

              There is considerable value in this ms. It looks at an important research question, uses an impressive and large national sample, and conducts appropriate analyses (I especially liked the use of cluster-robust standard errors, 463). The literature cited is voluminous as well (but, curiously, omits major papers and doesn’t seem to recognize the take-home points of some they do cite.) Nonetheless, I feel the paper needs substantial revisions to be acceptable.

Among my most important issues are that the authors seem not to appreciate the most important conclusion of the work: that in Germany, unlike in many nearby European countries, SPC is not “established by law” (67) nor has “institutional support for SPC” (105). As a result, SPC is notably infrequent and no contact with father cases are comparatively very high. This suggests that legal requirements AND institutional support are clear requirements for proliferation of SPC. Couples will only rarely adopt it when these are missing. If jurisdictions see value in SPC, as many do (e.g., Sweden, Netherlands, Australia, some U.S. states), they must either pass actual legislation that furthers it or their institutions must strongly encourage it. One can’t count on individual norms or personal agency to make it widespread. This seems to me a conclusion that should be emphasized in a revision.

Relatedly, the applicability of the findings for the two main research questions is totally limited to the context of a jurisdiction that doesn’t have legal or institutional support for SPC. Findings about which families adopt SPC (RQ1) and child adjustment outcomes for SPC (RQ2) are completely limited to jurisdictions that are similar in these respects—and in Europe, such jurisdictions are few. One might well expect quite different results for these two questions in dissimilar jurisdictions, where SPC is the rule rather than the exception. This limitation should be emphasized in a revision.

Also relatedly, Fabricius, Braver, Diaz, & Velez, C.E. (2010) (Custody and parenting time: Links to family relationships and well-being after divorce. In Michael E. Lamb (Ed). Role of the Father in Child Development (5th Edition). (pp. 245-289)) have found that of the two, institutional support is more important than legal presumptions. They note that there were substantial changes in the rates of SPC when divorce professionals changed their views and give different guidance to the families, even when the formal rules did not change.

Other points:

(361) The response rate of 21%, though respectable for a national study, should be mentioned as a limitation, since those who choose to participate are somehow unique and there is clearly selection bias. Some speculations should be made about how this bias might skew the portrait that emerges.

(384-391) Does the data set permit other definitions of SPC, occasional father contact, etc. than the arbitrary cutoffs used? If so, a much richer analysis (and a sensitivity analysis) can be undertaken by trying other cutpoints.

(411-416) Same comment about arbitrary operational definitions of distance between parental residences.

(417-436) It is a severe limitation that only mothers completed both the coparenting assessment and the SDQ. There is ample evidence that reporter effects condition findings like these. Father and child reports of these variables might well reveal different levels of problems as well as different associations with other variables. This is mentioned in the limitations (756-766) but should be amplified, with references that show what past research has shown about these various reports. Moreover, the contention that different findings for mother vs father report  are “particularly relevant” for “highly conflictual” families is questionable—again peruse the literature on this.

(496) I wish especially that results concerning differences between residence, the most pervasive predictor, had been broken down. Exactly, how much distance is it that precludes SPC and other variables?

(516) I am troubled by this study—and almost all others—treating coparenting as an exogenous variable. As the authors later note, coparenting itself has complicated cause and effect relationships with the other variables at issue in the study.

Author Response

Responses to Reviewer 1

  1. There is considerable value in this ms. It looks at an important research question, uses an impressive and large national sample, and conducts appropriate analyses (I especially liked the use of cluster-robust standard errors, 463). The literature cited is voluminous as well (but, curiously, omits major papers and doesn’t seem to recognize the take-home points of some they do cite.) Nonetheless, I feel the paper needs substantial revisions to be acceptable.

Among my most important issues are that the authors seem not to appreciate the most important conclusion of the work: that in Germany, unlike in many nearby European countries, SPC is not “established by law” (67) nor has “institutional support for SPC” (105). As a result, SPC is notably infrequent and no contact with father cases are comparatively very high. This suggests that legal requirements AND institutional support are clear requirements for proliferation of SPC. Couples will only rarely adopt it when these are missing. If jurisdictions see value in SPC, as many do (e.g., Sweden, Netherlands, Australia, some U.S. states), they must either pass actual legislation that furthers it or their institutions must strongly encourage it. One can’t count on individual norms or personal agency to make it widespread. This seems to me a conclusion that should be emphasized in a revision.

Relatedly, the applicability of the findings for the two main research questions is totally limited to the context of a jurisdiction that doesn’t have legal or institutional support for SPC. Findings about which families adopt SPC (RQ1) and child adjustment outcomes for SPC (RQ2) are completely limited to jurisdictions that are similar in these respects—and in Europe, such jurisdictions are few. One might well expect quite different results for these two questions in dissimilar jurisdictions, where SPC is the rule rather than the exception. This limitation should be emphasized in a revision.

Response: Thank you for these helpful suggestions! In response, we highlight and underscore the role of the study context (i.e., the lack of legal and institutional support for SPC in Germany to date) in shaping our findings and their applicability in various parts of the revised manuscript. For instance, we alert the reader to the fact that our results may only be applicable to similar jurisdictional contexts that also lack legal and institutional support for SPC in the introduction (Lines 69-73), discussion (Lines 634-640), and conclusion section (Lines 846-852) of the revised manuscript. We also clearly state that our research questions are focused on the German context only (Lines 333-363), which should further sensitize the reader to the fact that our findings may not be generalizable to any other context.

  1. Also relatedly, Fabricius, Braver, Diaz, & Velez, C.E. (2010) (Custody and parenting time: Links to family relationships and well-being after divorce. In Michael E. Lamb (Ed). Role of the Father in Child Development (5th Edition). (pp. 245-289)) have found that of the two, institutional support is more important than legal presumptions. They note that there were substantial changes in the rates of SPC when divorce professionals changed their views and give different guidance to the families, even when the formal rules did not change.

Response: Thank you for bringing this useful study to our attention, which we integrated into our discussion and conclusion section (e.g., Lines 676-681).

  1. (361) The response rate of 21%, though respectable for a national study, should be mentioned as a limitation, since those who choose to participate are somehow unique and there is clearly selection bias. Some speculations should be made about how this bias might skew the portrait that emerges.

Response: The reviewer is right in pointing out that an overall response rate of 21% could introduce non-response bias, even though lower response rates are not uncommon for large-scale nationally representative studies. However, a technical report (unfortunately only available in German; Kuger et al. 2021) that accompanies the dataset we used notes that there was no indication of severe self-selection bias for the overall sample with regard to basic socio-demographic characteristics compared to the German population.

            It is nevertheless true that our focus on respondents with an active coparenting relationship with the other parent likely resulted in some degree of self-selection bias. We acknowledge and discuss this limitation in the discussion section, as the editor requested (Lines 752-756).

  1. (384-391) Does the data set permit other definitions of SPC, occasional father contact, etc. than the arbitrary cutoffs used? If so, a much richer analysis (and a sensitivity analysis) can be undertaken by trying other cutpoints.

Response: We understand your concern! However, we would like to clarify that we did not use arbitrary cutoffs to define father-child contact because this indicator was measured as a categorical variable in the dataset we used. Respondents were asked to rate how often the other parent (i.e., usually the father) was in touch with the child. The response categories were: 1 = never, 2 = less than once a month, 3 = once or twice a month, 4 = once or twice a week, 5 = multiple times per week, 6 = daily. Based on this information and the count of overnight stays at the other parental residence per month, we then differenced our four post-separation physical custody arrangements (i.e., SPC [70:30 time split, which means least 10 and up to 21 nights per month at the other parental home] vs. sole care with either no [no overnight stays at the other parental residence and no father-child contact], occasional [less than 10 nights per month at the other parental residence and father-child contact was at least once or twice a month or less often], or frequent father-child contact [less than 10 nights per month at the other parental residence and father-child contact was at least once or twice a week or more often]). In the manuscript, we revised the description concerning the measurement of father-child contact to alert the reader to the fact that this was a categorical indicator (Lines 395-409).

            Nevertheless, it is true that the definitions of SPC vary widely in the literature and can range from an equal (50:50) to at least a one-third time split (70:30; e.g., Baude et al. 2016; Steinbach 2019). We chose to use the latter definition because of low case numbers for SPC, even when we use this rather unequal time split. As the reviewer rightly noted in a previous comment, the low case numbers for SPC are likely related to the lack of legal and institutional support for SPC in Germany compared to other countries where SPC was introduced as the legal default following a separation or divorce. In response to the reviewer comment, we reran our models using a more restrictive 60:40 time split and the patterns of results were largely in line with those based on the 70:30 time split, but did not always reach significance. More specifically, the distance between the parental residences was a significant predictor of SPC in the multinomial regression models, but the quality of the parental coparenting relationship did not reach significance. Relatedly, the pattern of results for the interaction between SPC and coparenting problems in the linear regression model predicting children’s level of hyperactivity was identical to our reported results, but was not significant at an alpha level of 0.05 (only at an alpha level of 0.10) using the 60:40 time split. This can most likely be attributed to a lack of statistical power—particularly for the detection of interaction effects—related to the very low case numbers for SPC when we use this more restrictive time split (max. N = 45 in the models without the parental coparenting relationship). Thus, we are reassured that the patterns of results also hold when we use the more restrictive 60:40 time split, even though some findings did not reach significance, and stand with our decision to use a 70:30 time split to define SPC, which is widely used in previous research on SPC (e.g., Sodermans et al. 2013; Recksiedler and Bernardi 2021).

  1. (411-416) Same comment about arbitrary operational definitions of distance between parental residences.

Response: This is an important point to clarify. In the dataset we used, the distance between the parental residences was measured with a categorical indicator. Participants were asked whether the residence of the other parent was 1 “in the same house,” 2 “in the same neighborhood,” 3 “in the same town or village, but more than 15 minutes away,” 4 “in a different town or village, but less than one hour away,” 5 “further away, but in Germany,” or 6 “further away, in another country.” We collapsed the first two categories (“in the same house” and “in the same neighborhood”) because of the low case number for the first category and then entered this indicator as a continuous variable (ranging from 1 “same house/neighborhood” to 5 “further away, in another country) into our models. This approach is common practice and recommended for ordinal indicators with at least five categories (e.g., Rhemtulla et al. 2012; Robitzsch 2020). As expected, our results revealed that if the parental residences were in closer proximity to each other, the chance that children practiced SPC was higher. Based on your comment, we also reran our models and entered the distance between the parental residence as a categorical indicator (reference category is same house/neighborhood; see Figure L1 below). It can be seen that there was no significant difference in children’s likelihood to practice SPC when the parental residences were in the same house/neighborhood vs. in the same town/village, but 15 minutes away. Children were more likely to be in SPC, however, when the parental residences were in the same house or neighborhood compared to those who reside in a different town/village, elsewhere in Germany, or abroad (both less than and more than an hour away). Relatedly, children were more likely to reside with their mothers and to have no father-child contact if the parental residences were in a different town/village, elsewhere in Germany, or abroad (both less and more than one hour away) compared to within the same house or neighborhood. These findings are in line with prior studies showing that SPC is only likely and feasible if the commutes between the parental homes are short (e.g., Kaspiew et al. 2009; Schier and Hubert 2015; Walper et al. 2021). In the manuscript, we revised the description concerning the measurement of the distance between the parental residences to alert the reader to the fact that this was a categorical indicator (Lines 429-440). We further summarize the pattern of results based on the categorical use of this indicator briefly (Lines 535-546).

Figure L1. Predicted probabilities of belonging to each post-separation physical custody arrangement based on socio-demographic and socio-economic predictors. Reference categories are: Female; Primary levels of education; Not employed; Parental residences in the same house/neighborhood; No economic deprivation; One child in the household.

  1. (417-436) It is a severe limitation that only mothers completed both the coparenting assessment and the SDQ. There is ample evidence that reporter effects condition findings like these. Father and child reports of these variables might well reveal different levels of problems as well as different associations with other variables. This is mentioned in the limitations (756-766) but should be amplified, with references that show what past research has shown about these various reports. Moreover, the contention that different findings for mother vs father report are “particularly relevant” for “highly conflictual” families is questionable—again peruse the literature on this.

Response: We agree with the reviewer that this is a valid concern and, in response, strengthened our discussion of potential reporter effects on our findings in the limitation section of our revised manuscript (Lines 818-828).

  1. (496) I wish especially that results concerning differences between residence, the most pervasive predictor, had been broken down. Exactly, how much distance is it that precludes SPC and other variables?

Response:  Our multinomial regression models predicting the likelihood of belonging to each respective post-separation physical custody arrangement (results are displayed in Figures 1 and 2 in the manuscript) address this question specifically. The predicted probabilities based on these models revealed that only the distance between the parental residences (see Figure 1) and the quality of the parental coparenting relationship (see Figure 2) were significantly associated with one’s likelihood to be in SPC.

First, if the parental residences (entered into the models as a continuous predictor) were in closer proximity to each other, SPC was more likely. As a sensitivity check, we also entered the distance between the parental residences as a categorical predictor into our models and the results showed that children were less likely to be in SPC if the parental residences were in a different town/village, elsewhere in Germany, or abroad (both less than and more than an hour away). For a more detailed overview of these results, please refer to our answer to your Comment 6 in this letter.

            Second, fewer problems in the parental coparenting relationship, which was measured on a 6-point Likert scale where higher values indicate more problems and were subsequently entered as a continuous predictor into our models, were associated with a higher likelihood to practice SPC. Relatedly, a more problematic coparenting relationship was predictive of sole custody arrangements with no or only occasional father-child contact.

  1. (516) I am troubled by this study—and almost all others—treating coparenting as an exogenous variable. As the authors later note, coparenting itself has complicated cause and effect relationships with the other variables at issue in the study.

Response: The reviewer is right in noting that the parental coparenting relationship is likely not an exogenous predictor of post-separation custody arrangements or child adjustment. Yet our study is limited to the use of cross-sectional data, which does not allow us to draw any causal claims or to examine these complex cause and effect relationships. We are upfront and clear about this limitation (Lines 803-810) and acknowledge the need for prospective studies to disentangle interdependencies and the directionality of effects between the choice of post-separation custody arrangements, parental conflict and coparenting, as well as child adjustment.

Reviewer 2 Report

            This is a straightforward and well-executed study of selection into shared parental custody and the association between shared parental custody and child well-being in a nationally representative sample of German children. An important and, surprisingly, somewhat distinctive element of the investigation is the inclusion of two key potential moderators of the relationships of interest, namely geographic proximity and co-parenting relationship quality. Key findings include (1) a low rate of shared parental custody in Germany relative to other countries in northern Europe, (2) evidence of nonrandom selection into shared parental custody based on geographic proximity, higher education/socioeconomic status, and a better co-parenting relationship, (3) no evidence of an association between shared versus sole parental custody and better child adjustment, and (4) some evidence supporting a moderating effect of co-parenting relationship quality such that, when co-parenting is worse, hyperactivity is higher among children in shared parental custody but lower among children in sole parental custody.

The study has limitations, including its cross-sectional design, definition of shared custody (defined as 30-70% time with both parents), and limited measurement of child well-being. However, the authors readily acknowledge these limitations, which also apply to almost all research on the topic. Indeed, the present study is stronger than most in its use of a nationally representative sample of children and reasonable large sample size. The report is also distinctive in its generally dispassionate and objective treatment of a topic that often is accompanied by strident advocacy, even in supposedly scientific reports.

            Turning to the study’s results, the low rate of shared custody in Germany has been documented previously. The evidence on nonrandom selection into shared custody also has been widely documented in studies throughout the world. The present replication is important, however, because nonrandom selection often is conveniently overlooked by those who wish to turn correlation into causation. Factors like a better co-parenting relationship predict both selection into shared custody and better child adjustment, providing a highly plausible alternative to the idea that shared custody causes any differences found in child adjustment in shared versus sole custody. Moreover, any correlation found between shared custody and better child adjustment often is either not statistically significant or of a small effect size, as again replicated in the present study.

            The most important contribution of the present research is the finding that shared parental custody is potentially harmful to children when parents fail to co-parent effectively. As a psychologist who has both practiced and conducted research in this area for decades, this finding is hardly surprising, yet this important moderating effect needs firmer empirical grounding (it is studied too rarely), especially since advocates of shared custody frequently argue against it. We learned 40 years ago that children fare better in a conflict-free single parent family than in a conflict-ridden two-parent family. It does not take a great leap of logic to expect that, in the face of co-parenting conflict, children also fare better in sole than shared custody. Like getting divorced versus staying married, sole versus shared custody provides far fewer opportunities for co-parenting conflict to be expressed in front of a child.

Logic and growing evidence, suggest that, when co-parenting is conflicted, children fare better in sole than shared custody. This creates a conundrum for policy makers and courts, one that, unfortunately, the present paper fails to recognize and discuss. State intervention is needed only when parents are in conflict. This means that policies designed to promote shared custody may target the right solution at the wrong group of parents.

There are two solutions to this conundrum. First, alternative dispute resolution such as divorce mediation holds the potential to both encourage better co-parenting, and as an outgrowth of parents working together, also encourage shared custody. In fact, I demonstrated that mediation can produce precisely these outcomes in my 12-year longitudinal study that began with randomized trials of the mediation and litigation of child custody disputes (citation withheld). Second, if parents cannot reach an agreement on their own, in mediation, or through their attorneys, judges need to be prepared to make difficult decisions, including awarding sole custody. Awarding shared custody may seem like a fair decision for disputing parents, but as biblical story of King Solomon warned, in such a circumstance, dividing the baby is not fair to children.

Author Response

Thank you for these insightful comments. The comparison with seminal work on the consequences of (a highly conflictual) parental divorce on child adjustment is interesting and very fitting. We incorporated some of these remarks into the discussion section of the revised manuscript (Lines 747-751). We also highlight the importance of mediation for conflictual parents in our conclusion section (Lines 863-869), which is likely to improve the parental coparenting relationship (Sbarra and Emery 2008). These efforts could subsequently help to foster post-separation child adjustment.
